# Analysis of the Elastoplastic Response in the Torsion Test Applied to a Cylindrical Sample

**DOI:** 10.3390/ma12193200

**Published:** 2019-09-29

**Authors:** Sebastián Andrés Toro, Pedro Miguel Aranda, Claudio Moisés García-Herrera, Diego Javier Celentano

**Affiliations:** 1Departamento de Ingeniería Mecánica, Universidad de Santiago de Chile, USACH. Av. Bernardo O’Higgins 3363, Santiago de Chile 9170022, Chile; sebastian.toroc@usach.cl (S.A.T.); pedro.aranda@usach.cl (P.M.A.); 2Departamento de Ciencia de Materiales, Universidad Politécnica de Madrid, E.T.S. de Ingenieros de Caminos, 28040 Madrid, Spain; 3Departamento de Ingeniería Mecánica y Metalúrgica, Pontificia Universidad Católica de Chile, PUC, Av. Vicuña Mackenna 4860, Santiago de Chile 8970117, Chile; dcelentano@ing.puc.cl

**Keywords:** torsion test, mechanical characterization, elastoplastic response

## Abstract

This work presents an experimental and numerical analysis of the mechanical behavior of a fixed-end SAE 1045 steel cylindrical specimen during the torsion test. To this end, an iterative numerical–experimental methodology is firstly proposed to assess the material response in the tensile test using a large strain elastoplasticity-based model solved in the context of the finite element method. Then, a 3D numerical simulation of the deformation process of the torsion test is tackled with this previously characterized model that proves to be able to predict the development of a high and localized triaxial stress and strain fields caused by the presence of high levels of angular deformation. Finally, the obtained numerical results are analytically studied with the cylindrical components of the Green–Lagrange strain tensor and experimentally validated with the measurements of shear strains via Digital Image Correlation (DIC) and the corresponding torque – twist angle curve.

## 1. Introduction

The mechanical characterization of the elastoplastic behavior of a material during the torsion test is relevant for the analysis and optimization of manufacturing processes that are nowadays carried out. One of the most elementary cases is the high resistance cables used in cranes, elevators, pulleys, etc. In the manufacturing of these elements, the material is subjected to extreme large torsion deformations, which in turn induce high elastoplastic stresses that complicate predicting the mechanical behavior of the material under service conditions.

In the case of pure torsion of cylindrical bars, it is well-known that the distribution of stresses is not uniform in the radial direction, leading to a zero value at the center of the specimen and to a maximum in its periphery, for both the elastic and elastoplastic regimes. Furthermore, some authors have reported the presence of axial deformations or axial forces, depending on the boundary condition used, when the cylindrical specimen is under a large torsional state. The most important case that occurs in the torsion test is the Swift effect [1], in which significant changes are generated in the original length of a cylindrical bar when it is subjected to large twist (i.e., torsion) angles.

In recent years, the study of torsion has been mainly focused on experiments to analyze the effect of large deformations on the material response. The large deformations induced in the torsion test were used to observe the changes in the microhardness on the crystalline structure [2]. Another experimental approach has been developed to analyze the changes in the mechanical response of the material by means of a treatment caused by hot-torsion [3], i.e., a test that is considered as the most common tool for studying the thermomechanical interaction during the hot rolling and radial casting processes.

Some authors [4,5] have proposed to analytically study the elastoplastic material response in the torsion test of a cylindrical bar considering both rate-independent and rate-dependent constitutive laws. More recently, numerical simulations using the finite element method have been reported to study the differences in the stress and strain patterns when adopting free or fixed ends to account for the Swift effect [6], the effect of the plastic spin in the macroscopic description of large deformation plasticity for the treatment of anisotropic hardening [7], the torsion flow curve compared with that observed in the tensile test [8], the tension–torsion high-cycle fatigue life prediction including anisotropic damage [9], the use of a model based on the corotational rates of the logarithmic strain and kinematic hardening to evaluate axial effects [10] and the analysis via polycrystalline plasticiy of the texture development and length changes in bars subjected to free-end torsion [11]. Although these studies had a solid mathematical basis, their experimental was in general limited.

Despite the torsion does not induce a necking zone in the specimen, as typically occurs in the tensile test, a complex stress state in the material is developed due to the zero shear stress condition present in the central fiber of the sample. This condition generates at high levels of deformation a steep gradient in the stress field. The literature available to analyse this problem is based on axisymmetric 2D models [8,9], making the quantification of 3D stresses such as those present in the Swift effect difficult. In spite of what has been mentioned, currently there are no experimental–numerical studies that analyze large deformations caused by pure torsion, considering a characterization based on the measurements obtained in the tensile tests, that can later be used to predict a process in which the material is subjected to large shear strains.

In the present work we analyze the mechanical response of the SAE 1045 steel in the torsion test using cylindrical samples. To this end, a twofold objective is pursued: to characterize the material hardening behavior through the tensile test and to carry out numerical 3D simulations of the torsion test by analyzing and validating the numerical results with the experimental curves. Therefore, this study encompasses experiments as well as numerical simulations. Uniaxial tensile tests have been firstly performed to calibrate, via an iterative experimental–numerical methodology that not only accounts for the engineering and true stress-strain curves but also for the ratio of current to initial diameter evolution at the necking zone, the material parameters of a large strain elastoplastic constitutive model based on the von Mises yield function including a Hollomon hardening law. Then, torsion tests with a fixed-end condition have been carried out where the previously characterized model is used for describing the material response under large angular deformations. In particular, the validity of some analytical expressions of the cylindrical components of the Green–Lagrange strain tensor is assessed by measuring the strain field using a 2D Digital Image Correlation (DIC). Moreover, the numerical prediction of the torque-torsion angle curve is experimentally validated while radial profiles of axial and shear stresses together with axial force evolution during the test are also computed and discussed. This manuscript is organized as follows. Section 2 describes the material and methods used, while the experimental and numerical results are presented and discussed in Section 3. Finally, the concluding remarks are drawn in Section 4.

## 2. Material and Methods

### 2.1. Experimental Procedure

#### 2.1.1. Material

The material used for the mechanical tests (tensile and torsion) corresponds to a commercial as-received SAE 1045 steel, whose average chemical composition is shown in Table 1, considering cylindrical specimens as sketched in Figure 1. A nearly linear gradual reduction in diameter is chosen in order to force the specimen fracture in the middle zone for both tests. This tapered profile fits the ASTM standards [12].

#### 2.1.2. Tensile Test

The tensile test is used here to establish the constitutive law of the material. Figure 2 shows the setup for the tests performed on a tensile machine. To check repeatability, 5 samples were used. A low load cell speed value of 2 mm/min was adopted to preclude rate-dependent effects (value within the range specified by the ASTM standards [12]). The number of measurements was 600 per minute The tests were carried out with an initial extensometer length $l_{0}=50$ mm, recording the force and displacement with precisions of 0.1 N and 1 μm, respectively. The load cell used in the testing machine has a maximum capacity of 100 kN. In addition, the external diameter evolution at the necking zone, whose initial value was $D_{0}=4.95$ mm, was recorded using a optical digital caliper with a precision of ±1 μm. The data acquisition was calibrated to gather 10 data per second.

In this test, the engineering stress and strain are respectively defined as $P/A_{0}$ and $\varepsilon =\left(l-l_{o}\right)/l_{o}$, where *P* is the recorded axial force, $A_{0}=\pi D_{0}^{2}/4$ is the initial transversal area and *l* is the instantaneous extensometer length measured during the test. As it is well known, the true stress and strain (logarithmic) are respectively defined as σ=P/A and $e=-2ln(D/D_{o})$, where $A=\pi D^{2}/4$ is the current transversal area at the necking zone with instantaneous diameter *D*. The following properties are obtained from this test: Young’s modulus, Poisson’s ratio and yield limit with the following ASTM standards 132-97 [13], 111-17 [14], 8M [12], respectively.

#### 2.1.3. Torsion Test

The torsion test provides the relationship between the torque and the angular deformation of the sample. It should be noted that larger homogeneous deformations are achieved in this case compared to those of the tensile test since no necking is developed in the sample. Figure 3 shows the setup of this test. Once again, 5 samples were considered. The angular velocity used was set to 1.5 rpm. The number of measurements was 600 per minute. The tests were carried out recording the torque and the angular motion of the jaws with a precision of 0.01 Nm and ±0.001°, respectively. The torque cell used in the testing machine has a maximum capacity of 45 Nm.

Additionally, a 2D Digital Image Correlation (DIC) has been carried out, in which the Lagrangian strain tensor of the central region of the specimen is obtained. The image correlation technique used considers a 2-D analysis that is comparable to the case of 3-D twisting under certain assumptions that have been corroborated. First of all the diameter of the specimen does not change during the test, which has been experimentally verified; see Figure 4. Secondly, the angular deformation considering a small axial deformation can be defined according to the expression: tan(γ)=(dϕ/dz)r=dS/dz where *r* is the radius of the specimen, γ is the shear deformation and dϕ is the angular variation in a cross section of width dz in the axial direction; see Figure 5. The term dS corresponds to the infinitesimal path of a point on the surface of the specimen. This path is perfectly circumferential under the assumption that the diameter of the specimen does not change during the test such that the path of a point on the surface of the specimen is given by the red arch.

The magnitude *h* can be measured directly with the DIC software. It should be noted that the displacement *Y* is the projection of the arch *S*. This displacement can be obtained using a 2D DIC analysis for any axial position. It can be shown that the displacement *Y* and the torsion angle (ϕ) can be geometrically related as: ϕ=acos(1−Y/r−h/r)−acos(1−h/r); S=rϕ. Therefore, using this equation, the DIC measurement in 2-D can be related to the angular path *S* for all the *z* positions visible in the video considered in the DIC analysis (see Figure 6). Thus, an indirect measurement of the shear deformation is obtained by numerical differentiation of *S* with respect to *z*, between points belonging to adjacent cross sections in areas of interest, such as the middle zone of the specimen where the deformation is concentrated. In this zone, the deformation measurement is averaged and correlated to a theoretical expression of the Green–Lagrange strain tensor for torsion problems (Equation (Equation 8) of the manuscript). The parameters that where employed in the DIC analysis are a temporal increment of 0.2 s/image, a space discretization of 3 × 3 pixels and a resolution of 0.04 mm/pixels. It should be noted that these measurements can be made up to torsion angles of about 70°.

### 2.2. Constitutive Modelling

The mechanical response of the selected material can be described by local governing equations expressed by the mass conservation, the balance of linear momentum and the dissipation inequality all described in a Lagrangian specification of motion. In this framework, the Cauchy non-polar stress tensor σ can be defined in terms of some thermodynamic state variables chosen in this work as the Almansi strain tensor e ($e=1/2(1-F^{-T}\cdot F^{-1})$, where F is the deformation gradient tensor and *T* is the transpose symbol) and a set of phenomenological internal variables (usually governed by rate equations with zero initial conditions) accounting for the non-reversible effects. The stress–strain relationship and the evolution laws for the internal variables adopted here to simulate the material behaviour are briefly described below (see [15] for further details).

In this work, the expression for the stress tensor σ (neglecting initial stresses) is given by:(1)$$\sigma =C:(e-e^{p})$$
where C is the isotropic elastic constitutive tensor and $e^{p}$ is the plastic Almansi strain tensor.

The chosen internal variables were $e^{p}$ and the effective plastic deformation ${\bar{e}}^{\;p}$. Their evolution equations are defined within the associate rate-independent plasticity theory context as:(2)$$L_{v}\left(e^{p}\right)=\dot{\lambda }\frac{\partial F}{\partial \sigma },\;{\dot{\bar{e}}}^{p}=-\dot{\lambda }\frac{\partial F}{\partial \sigma_{h}}$$
where $L_{v}$ is the well-known Lie (frame-indifferent) derivative, $\dot{\lambda }$ is the plastic consistency parameter, $\sigma_{h}$ is the isotropic hardening stress and $F(\sigma ,\sigma_{h})$ is the plastic flow potential. In this framework, *F* is also assumed as the yield function such that no plastic evolutions occur when F<0. In metal plasticity, *F* is usually chosen as the von Mises function:(3)$$F=\sqrt{3J_{2}}-\sigma_{h}-\sigma_{y}$$
where $J_{2}$ is the second invariant of the deviatoric part of σ ($\sigma_{eq}=\sqrt{3J_{2}}$ is the so-called equivalent or von Mises stress), $\sigma_{y}$ is the initial yield strength and $\sigma_{h}$ can be written according to the Hollomon law as:(4)$$\sigma_{h}=A^{p}{\left({\bar{e}}^{p}\right)}^{n^{p}}$$
where $A^{p}$ and $n^{p}$ are the parameters aimed at characterizing the hardening behaviour of the material. These hardening parameters can be directly obtained, as described in Section 3.1, through an experimental–numerical procedure.

The governing equations, together with the material constitutive model presented above, are discretized within the framework of the finite element method according to the numerical approach detailed in [15]. The computational implementation of the corresponding discretized equations is performed in an in-house code extensively validated in many engineering applications reported by the authors elsewhere. In this case, the equilibrium equation is solved in the material configuration where a B-bar type technique is used to avoid numerical locking due to plastic incompressibility. Furthermore, the integration of the plastic rate equation is carried out with the generalized midpoint rule algorithm choosing the parameter that makes the return mapping procedure unrestricted stable.

### 2.3. Numerical Modeling of the Tensile and Torsion Tests

The model briefly presented in Section 2.2 is used to simulate the tensile and torsion tests described in Section 2.1. These numerical analyses are necessary to achieve a proper interpretation of the material response due to the complex stress and strain patterns that develop in both tests. In particular, a necking formation under a triaxial stress with a non-homogeneous strain pattern occurs in the tensile test beyond the point of maximum load whereas a non-uniform axial stress distribution is developed in the torsion test for high levels of angular deformation. Different mesh sizes were analyzed until an element size was found to guarantee the convergence of the numerical results. The finite element meshes and boundary conditions considered in both tests are separately described below.

#### 2.3.1. Tensile Test

The spatially non-uniform finite element mesh shown in Figure 7 has been chosen in order to correctly describe the large stress and deformation gradients expected in the necking zone [15]. Assuming axisymmetry, a fourth of the specimen is discretized with a height of $l_{0}/2=25$ mm and, as mentioned in Section 2.1.2, a linear radius variation along the bar length. The domain is discretized with 868 quadrilateral elements with 944 nodes. An axial displacement, denoted as *U*, is imposed at the top boundary up to a value which corresponds to the experimental average fracture elongation.

#### 2.3.2. Torsion Test

The simulation of the torsion test was performed with a finite element mesh that consisted of 102,600 hexahedral elements with 107,679 nodes for one half of the cylindrical specimen including 30 mm of the hexagonal section that corresponds to the jaws coupling. First, the complete specimen was simulated with a fixed end and the other one applying the torsion angle. However, deformation symmetry was observed from the middle of the specimen. Therefore, only half of it was simulated by applying half of the torsion angle. The imposed boundary conditions and the mesh used are shown in Figure 8 where, according to the experimental conditions detailed in Section 2.1.3, a half of the twist angle together with a zero axial displacement were both prescribed on the hexagonal face.

### 2.4. Fitting Procedure for the Tensile Test

An initial guess of the hardening parameters $A^{p}$ and $n^{p}$ was obtained through the analytical procedure adopted in [15] that consists in the application of the least squares method on the equivalent stress–strain curve that results from the tensile test. To compare the results, we compute the experimental–numerical error as the normalized root-mean-square deviation (NRMSD) given by:(5)$$\mathrm{NRMSD}=\frac{1}{\Delta }\sqrt{\frac{1}{n}\sum_{i=1}^{n}{\left({\hat{y}}_{i}-y_{i}\right)}^{2}}$$
where *n* represents the number of registered values (it is considered the same for all samples), $y_{i}$ is the average experimental measurement, for all tested samples ${\hat{y}}_{i}$ is the numerical fitted value and $\Delta =|y_{max}-y_{min}|$. Since the NRMSD value associated to the parameters of the initial guess was relatively high (i.e., 30%), another strategy to derive reliable hardening parameters leading to a more accurate modeling of the material response must be defined [16].

For this purpose, the methodology consists in extending the previous approach to simultaneously account for three curves: (1) axial engineering stress–strain, (2) axial true stress–strain, and (3) ratio of current to initial diameter at the necking zone versus axial elongation. These curves are computed via a numerical simulation of the tensile test since, as already commented, this kind of analysis is needed to properly describe the complex stress and strain patterns that develop during the elongation process. The full iterative fitting algorithm is outlined in Figure 9.

With the initial guess of the hardening parameters $A^{p}$ and $n^{p}$, the following step is the computation, via the results provided by the simulation of the tensile test, of the NRMSD for the three above-mentioned curves. The hardening parameters are iteratively modified until a maximum admissible error of 5% is reached simultaneously for these three calibration curves through a standard optimization gradient algorithm.

## 3. Results and Discussion

### 3.1. Tensile Test

Figure 10, Figure 11 and Figure 12 respectively present the experimental and numerical results of the engineering stress–strain, true stress–strain, and ratio of current to initial diameter at the necking zone in terms of the axial elongation. The numerical curves correspond to the predictions obtained with the final hardening parameters summarized in Table 2 derived from the application of the iterative procedure outlined in Figure 9 (it should be noted that the values of *E*, ν, and $\sigma_{y}$ have also been directly obtained from the tensile test measurements, where *E*
$\sigma_{y}$ correspond to the stress in a 0.2% level of deformation). The vertical bars indicate the standard deviation where a low dispersion of the experimental measurements can clearly be appreciated (i.e., less than 3 % in each curve). In particular, the $D/D_{0}-\left(L-L_{0}\right)/L_{0}$ curve was found to be the most sensible to the variation of $A^{p}$ and $n^{p}$ due to the deformation process is mainly developed in the postnecking range, starting from an elongation value of 1.5% (that corresponds to the engineering strain the ultimate tensile strength clearly seen in Figure 10) up to the fracture stage. Overall, it is seen that this fitting procedure provided a good description of the mechanical response during the whole test.

### 3.2. Torsion Test

The analytical expression for the deformation gradient tensor at the periphery (edge) of the specimen can be written as [17]:(6)$$F=\left[\begin{array}{ccc} 1 & 0 & 0\\ 0 & 1 & \gamma \\ 0 & 0 & 1 \end{array}\right]$$
where γ is the angular distortion that varies linearly with the torsion angle [17]. Using this relation, the Green–Lagrange (at first, this tensor is defined as $E=1/2(F^{T}\cdot F-1)$) and Almansi strain tensors are respectively given by:(7)$$E=\left[\begin{array}{ccc} 0 & 0 & 0\\ 0 & 0 & \gamma /2\\ 0 & \gamma /2 & \gamma^{2}/2 \end{array}\right]$$
(8)$$e=\left[\begin{array}{ccc} 0 & 0 & 0\\ 0 & 0 & \gamma /2\\ 0 & \gamma /2 & -\gamma^{2}/2 \end{array}\right]$$
i.e., the non-zero components are $E_{\theta Z}=e_{\theta Z}=\gamma /2$ and $E_{ZZ}=-e_{ZZ}=\gamma^{2}/2$. The evolutions of these Lagrangian components in terms of the torsion angle are shown in Figure 13 together with the corresponding DIC measurements and numerical results obtained with the constitutive model described in Section 2.2. It is seen that the simulated shear component of the strain exhibits, in contrast to the analytical prediction, a nonlinear response. A significant difference between numerical and analytical models also occurs for the axial component of the strain. The two models give similar values for both strain components up to a torsion angle of 20°. Moreover, the numerical results reasonably agree with the DIC measurements for both the shear and axial strain components. For the DIC results, the noise is caused the numerical differentiation method described in the Section 2.

Although the simulated $E_{\theta Z}$–torsion angle curve is, as already commented above, non linear, the analytical relationship $E_{ZZ}=2E_{\theta Z}^{2}$ is however fulfilled, as shown in Figure 14a. Furthermore, considering a monotonic deformation process and negligible elastic strains, the accumulated effective plastic deformation can be obtained (using standard tensor notation) as [18]:(9)$${\bar{e}}^{p}=\sqrt{\frac{2}{3}e:e}=\frac{\gamma }{\sqrt{3}}\sqrt{1+\frac{\gamma^{2}}{2}}\approx \frac{\gamma }{\sqrt{3}}$$

Figure 14b shows that the analytical expression $\gamma =2E_{\theta Z}/\sqrt{3}$ is also verified with $E_{\theta Z}$ obtained from the simulation. These results confirms that the elastic part of the deformation is negligible compared to the plastic part for large torsion angles.

Figure 15 compares the experimental and computed results (using the parameters listed in Table 2) of the torque in terms of the imposed torsion angle ϕ. Both a low deviation in the experimental measurements and a good numerical prediction can clearly be seen. The main difference between these curves appear at the beginning of the process when the material starts hardening; this effect is more pronounced in the experiments than in the simulation. Moreover, the equivalent stress at the periphery of the sample for the torque value of 14 Nm approximately corresponds, according to the von Mises criterion, to $\sigma_{y}/\sqrt{3}$ i.e., from this loading state the plastic behavior starts to develop.

The computed radial profiles for the shear stress at the middle cross section of the sample are plotted in Figure 16 for different torsion angles. For a torsion angle of *ϕ* = 20°, a linearly elastic behavior occurs up to almost $0.5R/R_{o}$; from there to the periphery of the sample, an elastoplastic response develops with a relatively smooth hardening. For the cases with *ϕ* = 120° and *ϕ* = 350°, the steep gradient in the center reflects the narrowing of the elastic region whereas the elastoplastic zone evolves similarly to the case with a small torsion angle. Although the numerical singularity close to the coordinate R=0 is, as reported in [8], a challenging aspect in the simulation, note that it is adequately reproduced by the present model. Moreover, the stress distributions at any plane parallel to plane 1 along the working zone of the sample are found to follow the same trend as those shown in Figure 16.

The development of longitudinal stress has been previously analyzed [6,7,10], reporting the existence of compression forces in fixed-end torsion tests. This phenomenon is also assessed here through the computed radial profiles for the longitudinal stress at the middle cross section of the sample for different torsion angles shown in Figure 17. Although low levels of $\sigma_{zz}$ develop for *ϕ* = 20° and *ϕ* = 120°, this stress exhibits a noticeable profile for *ϕ* = 350°.

At $R/R_{o}=0.1$, the effect of compression and shear are relevant for *ϕ* = 350°, making the stress state clearly triaxial. Once again, this trend is also repeated at any plane parallel to plane 1 along the working zone of the sample.

The axial component of the stress has a nonlinear behavior with respect to the torsion angle as seen in the Figure 18 at the center and edge of the sample. It is observed that in the periphery of the sample, the axial stress is predominant up to approximately *ϕ* = 130° retaining a linear evolution after *ϕ* = 75°, while $\sigma_{zz}$ at the center monotonically increases.

Figure 19 shows the compressive axial force $F_{z}$ that results from the area integration of $\sigma_{zz}$ as a function of the torsion angle. Three stages can clearly be seen in the force evolution: almost negligible values for low angles due to plastic initiation only at the periphery of the sample (i.e., up to *ϕ* = 10°), rapid increase due to the radial progress of plastic effects with relatively high hardening evolution (from *ϕ* = 10° to *ϕ* = 75°) and continuous increase once almost the entire transversal section linearly hardens more smoothly (from *ϕ* = 75° to *ϕ* = 350°); see Figure 19. It should be noted that the highly non-uniform $\sigma_{zz}$ distribution shown in Figure 17 cannot be appreciated in the $F_{z}$ curve due to the central localized character of the longitudinal stress, so the behavior of the axial force is quite similar to that of $\sigma_{zz}$ at the edge as shown in Figure 18.

## 4. Conclusions

The elastoplastic response in the torsion test of the SAE 1045 steel has been characterized considering an associated isotropic constitutive model. Based on the tensile test measurements, an iterative numerical-experimental methodology was firstly used to fit the material hardening parameters. Then, this calibration procedure was satisfactorily validated in the simulation of the mechanical behaviour of the material in the torsion test where realistic predictions were obtained for the whole studied deformation levels. In particular, the shear and axial stresses were evaluated in the plastic regime. For high torsion angles (close to the material rupture stage), an important stress localization was observed in the central area of the specimen, an effect that makes numerical modeling necessary for its interpretation. Finally, relevant effects not included in the present study, such as kinematic hardening and damage evolution, could be addressed in future research using this test under cyclic conditions. It is known that due to warping torsion for high levels of deformation, some effects can appear, such as an asymmetric elastoplastic behavior, restraints due to boundary conditions, and noncircular section along the specimen length. These effects have not been taken into account in this study.

## Figures and Tables

**Figure 1 materials-12-03200-f001:**
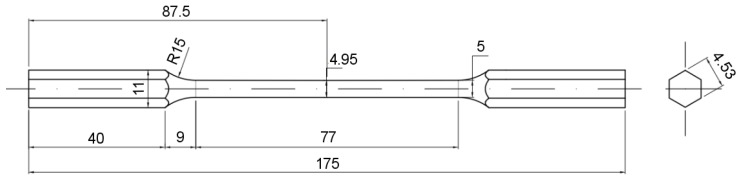
Geometric configuration of the SAE 1045 steel tensile/torsion cylindrical specimen (dimensions in mm).

**Figure 2 materials-12-03200-f002:**
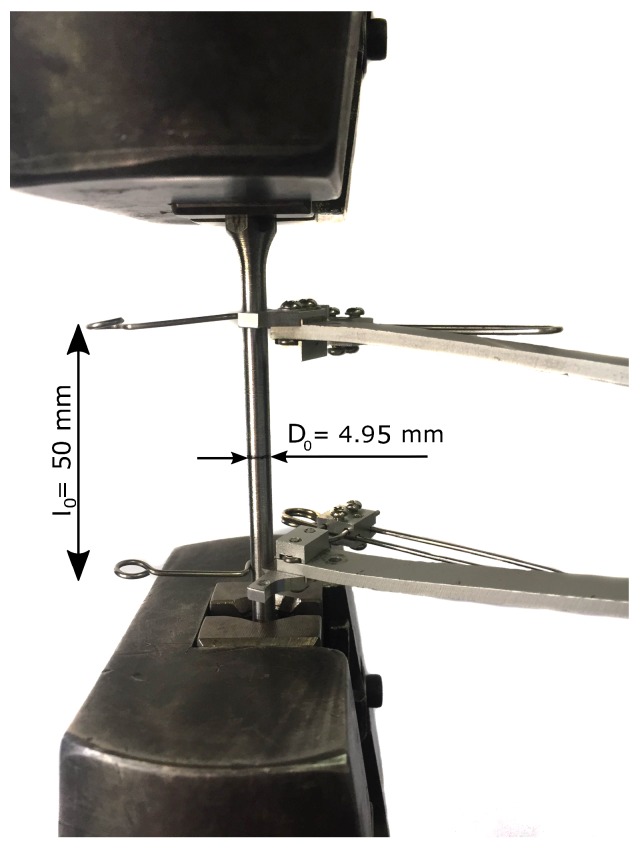
Setup of the tensile test.

**Figure 3 materials-12-03200-f003:**
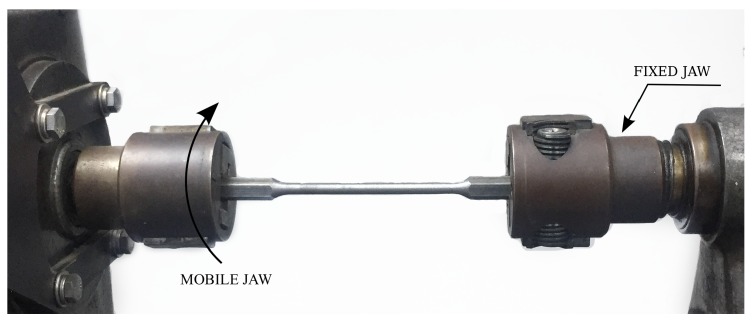
Setup of the torsion test.

**Figure 4 materials-12-03200-f004:**
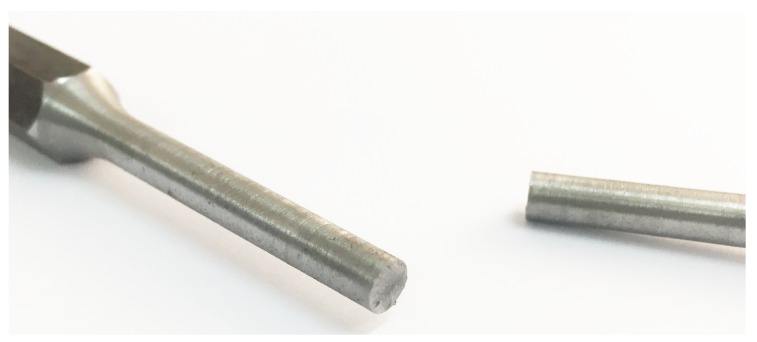
Failure condition of the torsion sample.

**Figure 5 materials-12-03200-f005:**
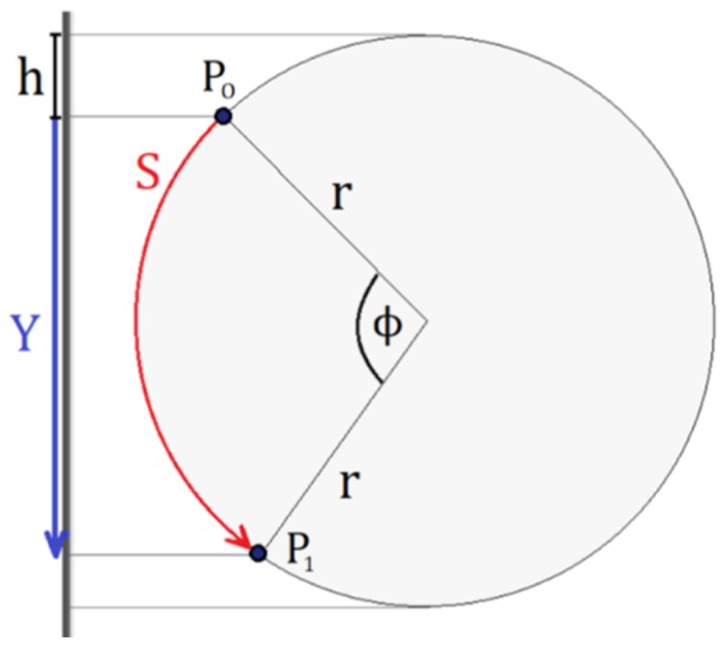
Kinematics of the torsion test for 2D Digital Image Correlation (DIC) measurements.

**Figure 6 materials-12-03200-f006:**
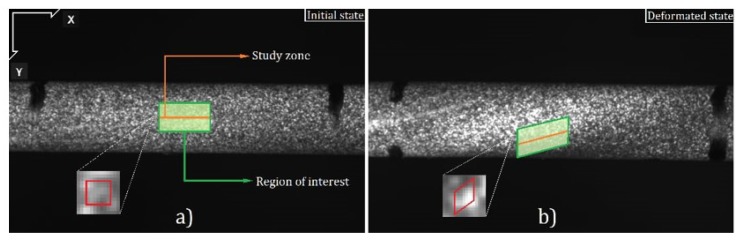
(**a**) Initial and (**b**) deformed DIC images.

**Figure 7 materials-12-03200-f007:**
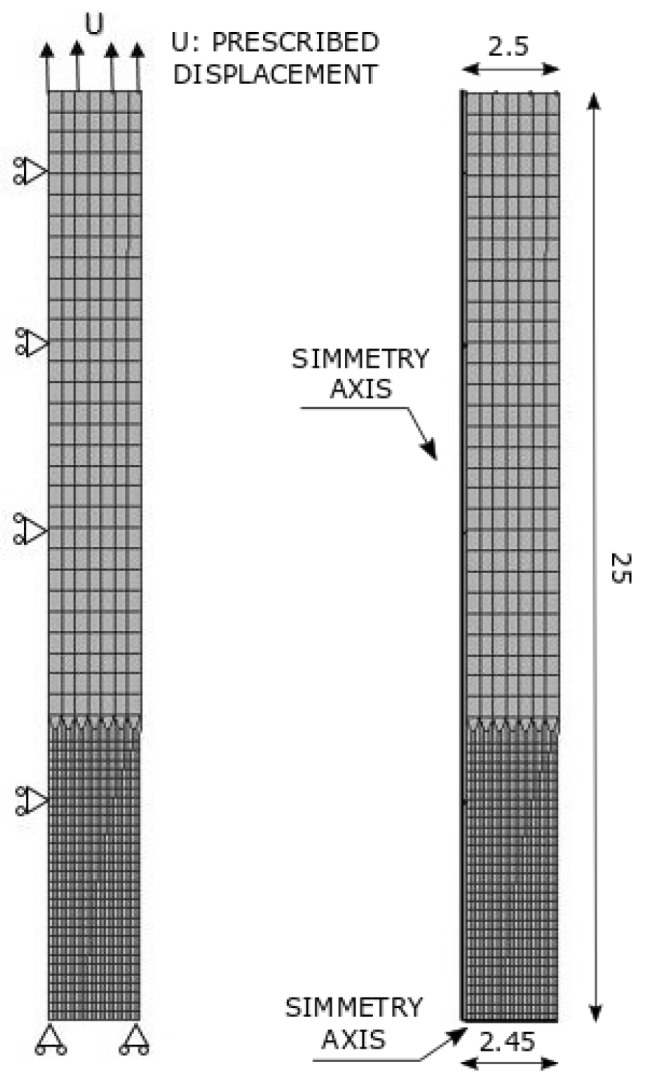
Tension test: finite element mesh and boundary conditions (dimensions in mm).

**Figure 8 materials-12-03200-f008:**
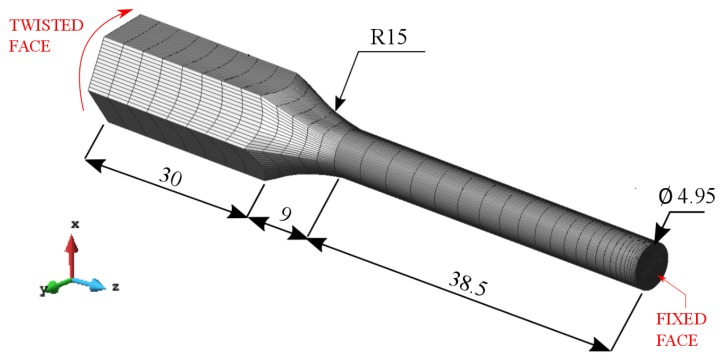
Torsion test: finite element mesh and boundary conditions (dimensions in mm).

**Figure 9 materials-12-03200-f009:**
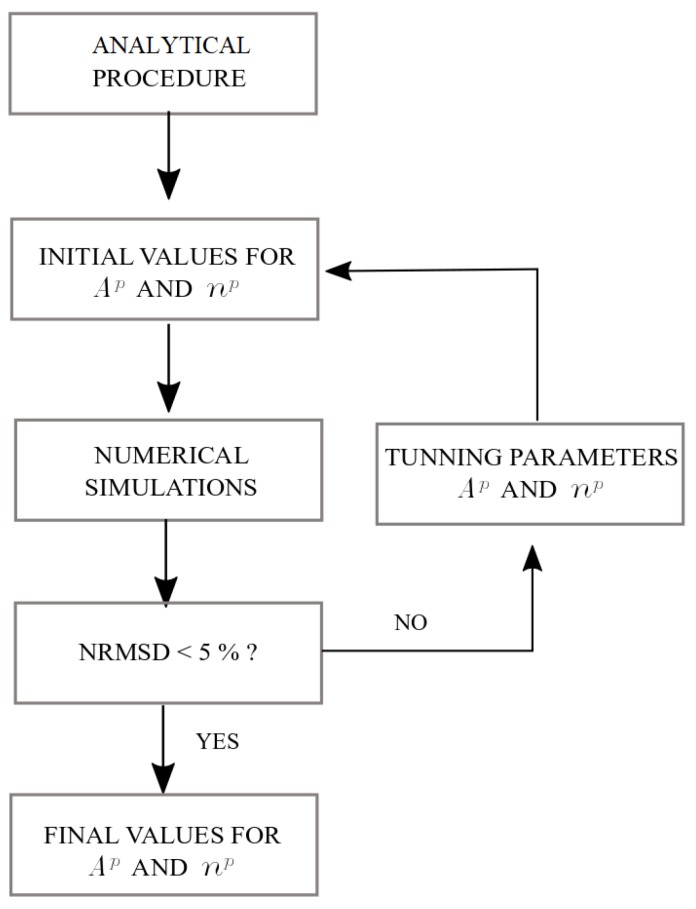
Iterative experimental–numerical procedure to obtain the hardening parameters $A^{p}$ and $n^{p}$.

**Figure 10 materials-12-03200-f010:**
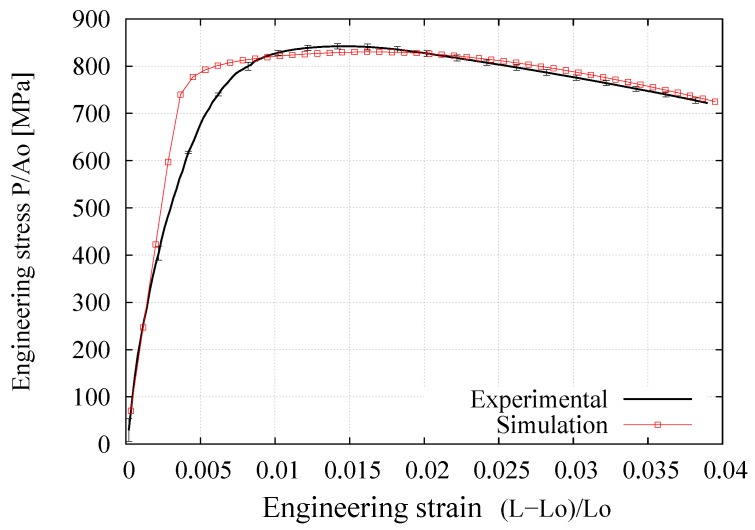
Average experimental and numerical results for the engineering stress–strain relationship, normalized root-mean-square deviation (NRMSD) = 4.67%.

**Figure 11 materials-12-03200-f011:**
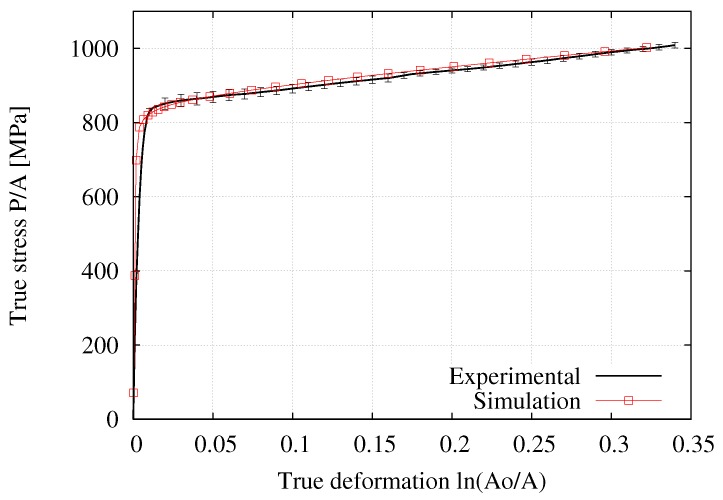
Average experimental and numerical results for the true stress–strain relationship, NRMSD = 0.91%.

**Figure 12 materials-12-03200-f012:**
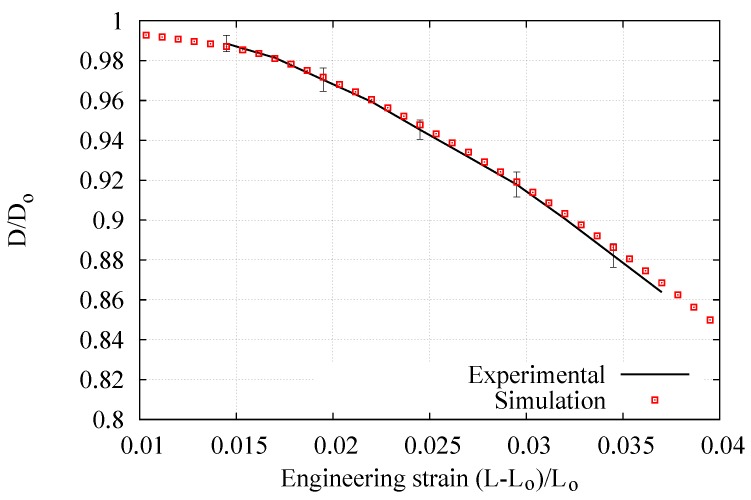
Average experimental and numerical results for the ratio of current to initial diameter at the necking zone versus axial elongation, NRMSD = 1.43%.

**Figure 13 materials-12-03200-f013:**
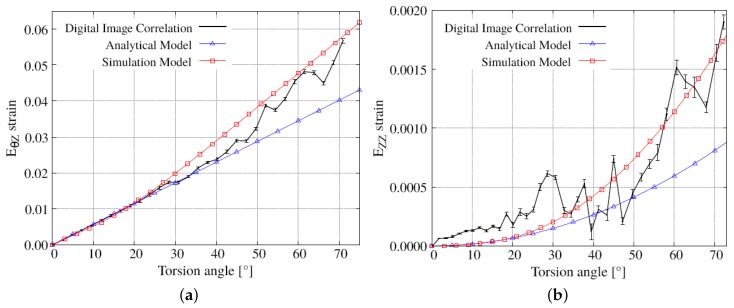
Average experimental, analytical, and numerical results for the strain components (**a**) $E_{\theta Z}$, NRMSD = 6.5% and (**b**) $E_{ZZ}$, NRMSD = 8.6%, expressed as a function of the torsion angle at the edge of the sample.

**Figure 14 materials-12-03200-f014:**
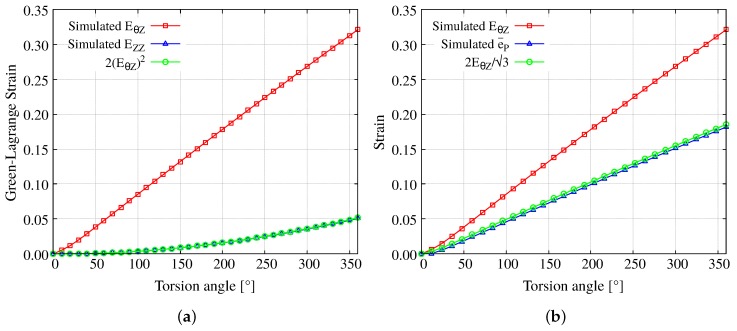
Verification of the analytical expressions for (**a**) $E_{ZZ}$ and (**b**) ${\bar{e}}^{p}$ using $E_{\theta Z}$ obtained from the simulation.

**Figure 15 materials-12-03200-f015:**
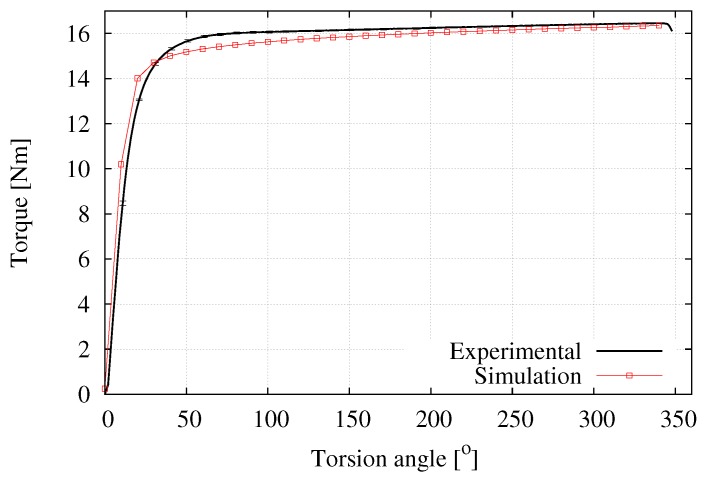
Average experimental and numerical results for the torque as a function of the torsion angle, NRMSD = 2.26%.

**Figure 16 materials-12-03200-f016:**
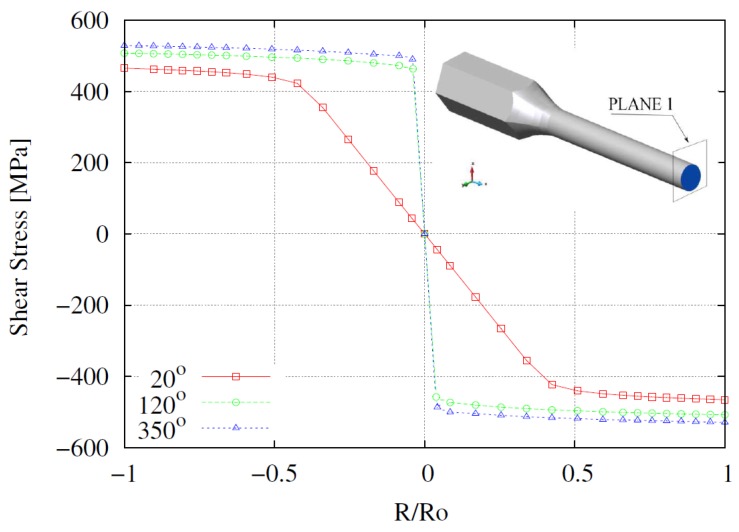
Numerical radial profiles for the shear stress $\sigma_{z\theta }$ in plane 1 for different torsion angles.

**Figure 17 materials-12-03200-f017:**
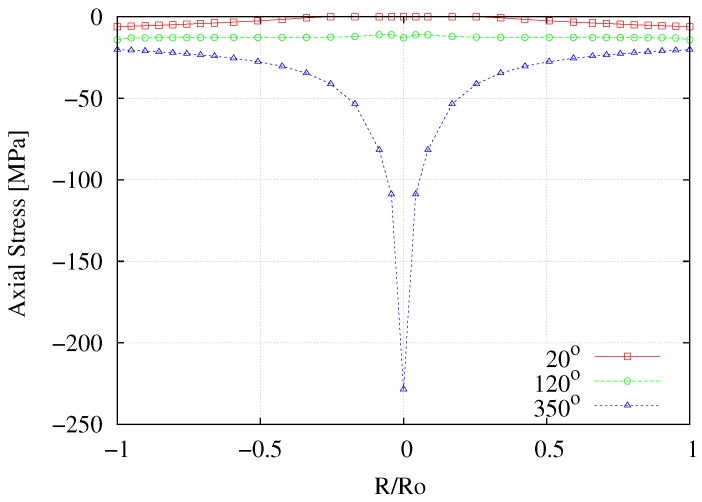
Numerical radial profiles for the longitudinal stress $\sigma_{zz}$ in plane 1 for different torsion angles.

**Figure 18 materials-12-03200-f018:**
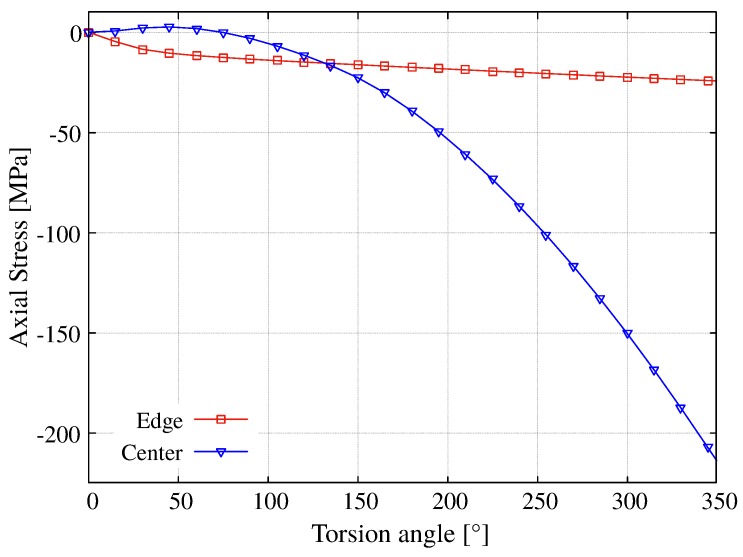
Numerical radial profiles for the longitudinal stress $\sigma_{zz}$ in plane 1 for different angular deformations.

**Figure 19 materials-12-03200-f019:**
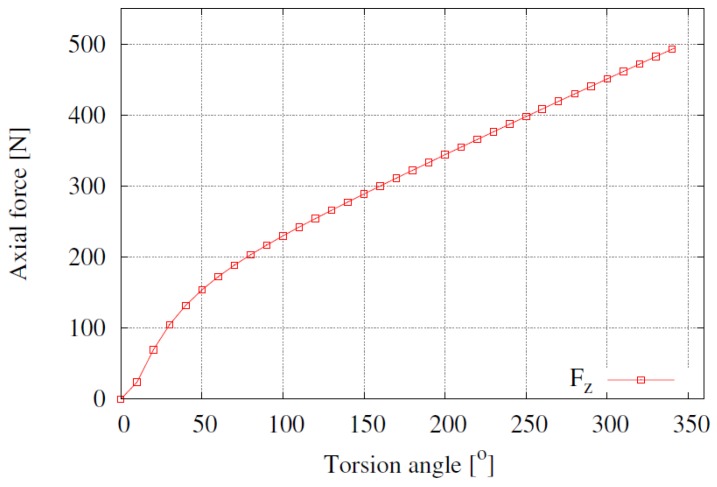
Numerical results for the axial compression load as a function of the torsion angle.

**Table 1 materials-12-03200-t001:** Average chemical composition of SAE 1045 steel (% in weight).

C [%] 0.433	Si [%] 0.218	Mn [%] 0.73	P [%] 0.01	S [%] 0.013
Cr [%] 0.019	Mo [%] 0.014	Ni [%] 0.044	Al [%] 0.0023	Cu [%] 0.042
Nb [%] <0.001	Ti [%] 0.0009	V [%] 0.0022	W [%] <0.007	Pb [%] <0.001
Sn [%] 0.0059	B [%] 0.0027	Co [%] 0.0088	Fe [%] 98.3	

**Table 2 materials-12-03200-t002:** Derived material parameters.

Parameter	Value
E	208 GPa
ν	0.271
Tensile strength	840 MPa
$\sigma_{y}$	560 MPa
$n^{p}$	0.033
$A^{p}$	950 MPa

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
