# Peer review of "Analysis of the Elastoplastic Response in the Torsion Test Applied to a Cylindrical Sample"

_materials, 2019, doi:10.3390/ma12193200_

Round 1

Reviewer 1 Report

The paper deals with an experimental and numerical analysis of the mechanical behavior of a fixed-end SAE 1045 steel cylindrical specimen during torsion tests. The aim of the work is of interest to the journal readers but it needs to be completeley reorganized, deeper analyzed and better written and discussed. Some parts of the paper are unclear.

 It is no clear if the curves of experimental results, used for comparisons, are mid-curves or specific curves. Parameters of DIC are not mentioned (resolution, etc...)

Anyway, The principal problem is given by the coomparison between 3D FE results and 2 D DIC analyses. The authors state "a 3D numerical simulation of the deformation process of the torsion test is tackled with this previously characterized model that proves to be able to predict the development of a high and localized triaxial stress and strain fields caused by the presence of high levels of angular deformation". The problem is that 2D DIC was used, while, having 3D displacements, a 3D technique would be necessary. 

Boundary conditions of figure 6 are different compared to the experimental ones. There is no angle equal to zero in the meddle of the length of the specimen in reality. At which node and position were the results evaluated for comparisons with experimental tests?

For these reasons, the reviewer suggests a deeper and more functional analysis, a complete reorganization and a new submission.

Reviewer 2 Report

The submitted article presents an experimental and numerical study on the torsional behaviour of SAE 1045 steel cylinder specimens. Firstly, a methodology based on uniaxial tensile tests, both experimentally and numerically by using finite element method (FEM), is proposed to establish the constitutive law of the material, including the large strain range. Then, experimental torsional tests are carried out, as well as numerical simulations with FEM based on the previous law, in order to predict the full torsional behaviour of the specimens, including the triaxial stress and strain fields which arise for large angular deformations and elastoplastic behaviour. The results are presented and discussed. Good agreement was observed between the experimental and numerical results.

The topic that is developed in the article is very interesting and still needs further research. The obtained results can be useful for engineers and industry dealing with the manufacture of steel components.

I made few specific comments (see below) to improve the article. The reviewer encourages the authors to take the few suggestions into account and resubmit a reviewed version of the article.

Comment 1

The article does not respect the instructions for authors. The template provided in the MDPI Materials site must be used. 

Comment 2

Some typos must be corrected. For instance, in page 19 and 4th line from the bottom, I think that “see Figure 14” should be “see Figure 17”. Also, in page 18 and 3th line from the bottom, please delete the “i” next to “fi = 130º”

Comment 3

A note about the possible effects due to warping torsion should be added. In an ideal steel cylindrical specimen under elastic torsion, warping torsion don´t exist. However, in the used test specimens, some effect can exist due to many reasons, such as: asymmetric elastoplastic behavior, restraints due to boundary conditions, noncircular section and nonuniform section along the specimen length, …

Comment 4

It would be interesting to include the relevant mechanical properties of the SAE 1045 steel provided by the supplier and compare them with the experimental and/or numerical results.

Comment 5

A didn´t find information about the used FEM software. This information should be given.

Reviewer 3 Report

See attached file

Reviewer 4 Report

The authors performed experimental and numerical studies of the mechanical response of a cylindrical steel sample under torsion. The research of the paper is within the scope of the Journal with detailed experimental test and numerical modeling description, results, and mutal correlations. The paper can be considered for publication in this Journal provided that a few minor issues can be further addressed.

(1) Use of the Eq. (1) for material constitutive law needs to be addressed in detail as it is the core of the study.

(2) If available, a couple of images to show the necking stage of the samples or the failure cross sections can be helpful and enhance the paper work.

(3) It is a little unusual, the engineering stress-strain diagram in Figure 8 does not show the linear and yield range of the material. It seems a high-strength alloy. In this case, the authors may give the 0.2% offset yield stress as reference.

(4) The authors are suggested to correlate the tensile failure to the shear failure via classic failure theories.

(5) The authors are suggested to clarify the oscillations (noises) observed in Figure 11 (torsional strain components), just because of numerical differentiation?  

Round 2

Reviewer 1 Report

The authors have significantly improved the manuscript and properly answered the questions. The use of 2d DIC, applying simplification hypothesis, has been now motivated also in the manuscript and the paper can be accepted for publication.

Reviewer 2 Report

I received and read the revised version of the manuscript “Analysis of the elastoplastic response in the torsion test applied to a cylindrical sample”. I´m generally satisfied with the author´s replies to my earlier comments and I also consider that all my suggestions and concerns have been properly explained and considered by the author to improve the manuscript.

I consider that the revised manuscript submitted by the author can be accepted in the present form to be published.